# Dinitroimidazoles as bifunctional bioconjugation reagents for protein functionalization and peptide macrocyclization

Qunfeng Luo[1], Youqi Tao[1], Wangjian Sheng[1], Jingxia Lu[1] & Huan Wang[1]

Efficient and site-specific chemical modification of proteins under physiological conditions remains a challenge. Here we report that 1,4-dinitroimidazoles are highly efficient bifunctional bioconjugation reagents for protein functionalization and peptide macrocyclization. Under acidic to neutral aqueous conditions, 1,4-dinitroimidazoles react specifically with cysteines via a *cine*-substitution mechanism, providing rapid, stable and chemoselective protein bioconjugation. On the other hand, although unreactive towards amine groups under neutral aqueous conditions, 1,4-dinitroimidazoles react with lysines in organic solvents in the presence of base through a ring-opening & ring-close mechanism. The resulting cysteine- and lysine-(4-nitroimidazole) linkages exhibit stability superior to that of commonly employed maleimide-thiol conjugates. We demonstrate that 1,4-dinitroimidazoles can be applied in site-specific protein bioconjugation with functionalities such as fluorophores and bioactive peptides. Furthermore, a bisfunctional 1,4-dinitroimidazole derivative provides facile access to peptide macrocycles by crosslinking a pair of cysteine or lysine residues, including bicyclic peptides of complex architectures through highly controlled consecutive peptide macrocyclization.

[1] State Key Laboratory of Coordination Chemistry, Jiangsu Key Laboratory of Advanced Organic Materials, School of Chemistry and Chemical Engineering, Nanjing University, 210093 Nanjing, China. Correspondence and requests for materials should be addressed to H.W. (email: wanghuan@nju.edu.cn)

Chemical modification of proteins has the potential to expand their function by mimicking naturally occurring post-translational modifications[1–3]. Methods for site-specific modification of proteins are invaluable in generating biological therapeutic agents, such as antibody-drug conjugates (ADCs) and PEGylated biologics[3]. Although biorthogonal methods using engineered amino acids provide an elegant strategy for protein functionalization in complex biological mixtures[4], chemical means to prepare homogeneous proteins with defined modification in sufficient quantities are still required. Amongst all natural amino acids, cysteines are the most important amino acid for protein modification due to their intrinsic nucleophilicity[3,5]. Methodologies for cysteine modification often rely on electrophilic reagents such as maleimides (Michael acceptors) and aryl and alkyl halides[3,6]. Recently, a number of strategies have been developed to expand the toolbox for cysteine modification, including transition metal-mediated bioconjugation[7], a cysteine-to-dehydroalanine conversion strategy[8,9], enzymatic click cysteine ligation[10], and strain-release alkylation[11], among others[5,6,12–14]. Notwithstanding their advantages for bioconjugation, cysteines are rare in human proteins (1.9% of residues) and often exist as disulfides, and therefore unavailable for modification. Lysine residues, on the other hand, occur much more commonly in proteins (5.9% of residues in human proteins) and are generally exposed on the protein surface, providing a convenient target for bioconjugation. Typical lysine bioconjugation methods also rely on nucleophilic addition by lysine ε-amine, such as acylation, activated esters[15,16], and the recently developed arylation and condensation reactions[17–20]. Despite recent advances in Cys and Lys bioconjugation, chemoselective modifications are often difficult to achieve in a proteinaceous environment. For example, maleimides are the most widely used cysteine bioconjugation reagents[21]. However, maleimides often react with N-terminal amine and lysine under neutral pH, resulting in a mixture of Cys and Lys adducts (Fig. 1a)[14]. Compound TAK-242 is employed as a covalent Lys inhibitor to modify Lys64 of human serum albumin selectively; however, TAK-242 was found to react with Cys747 of Toll-like receptor-4 via the same nucleophilic addition mechanism (Fig. 1a). Moreover, the cross-reactivity of these bioconjugation reagents always causes the same mass change to target proteins, posing additional challenges for comprehensive proteomic studies. Although a certain degree of chemoselectivity between Cys and Lys can be achieved by careful control of reaction conditions (such as fine-tuning the pH of the media), cross-reactivity remains a major challenge for most bioconjugation reagents.

The stability of protein conjugates is another key criterion for bioconjugation reagents. Maleimides are the most widely used cysteine bioconjugation reagents, and are currently employed in most of the ADCs in the market and in clinical trials[21]. However, the susceptibility of the thiosuccinimidyl linkage to hydrolysis and thiol exchange reactions has limited its application in biological systems[22–24]. Efforts to overcome this limitation have been made by developing maleimide derivatives that lead to stable bioconjugates, such as the exocyclic olefinic maleimides, pyridazinediones, and hydrolysable maleimide derivatives[24–29]. Therefore, alternative bioconjugation reagents with high reactivity and selectivity are still in need to generate stable protein conjugates for pharmaceutical applications. Herein, we report that 1,4-dinitroimidazoles (1,4-DNIms) are a class of rapid, chemoselective, and bifunctional bioconjugation reagents for protein functionalization and peptide macrocyclization. 1,4-DNIms are cysteine-specific under a wide pH range (pH 3.0–8.0) in aqueous solutions, and modify cysteine residues in proteins quantitatively in seconds through a cine-substitution mechanism (Fig. 1b). Although unreactive towards lysines under neutral aqueous conditions, 1,4-DNIms modify lysine residues efficiently in organic solvents, such as dimethyl sulfoxide (DMSO), with weak bases through an ring-opening and ring-closing mechanism for protein bioconjugation. The resulting cysteine- and lysine-(4-nitroimidazole) linkages are significantly more stable than the commonly used maleimide conjugates. Using this chemistry, we have achieved cysteine-specific functionalization of proteins with fluorophores and bioactive peptides. In addition, we demonstrate that bisfunctional 1,4-DNIm compounds can provide facile access to peptide macrocycles with complex ring structures.

## Results

**1,4-DNIms are rapid and Cys-specific bioconjugation reagents.** To initiate our investigation, we synthesized compound **1a** and sought to evaluate its potential for cysteine bioconjugation (Fig. 2a). Incubation of compound **1a** (1.0 mM) with N-acetyl-L-cysteine (1.0 mM) in 100 mM HEPES buffer at pH 8.0 leads to rapid and quantitative conversion to a cysteine adduct **1b** in 15 s (Fig. 2a). Structural characterization of adduct **1b** by nuclear magnetic resonance (NMR) and X-ray crystallographic analysis indicate that the cysteine thiol is directly conjugated to C-5 of the nitroimidazole (Fig. 2a, Supplementary Fig. 1). This result suggests that the reaction proceeds through a cine-substitution mechanism with N1-nitro as the leaving group upon nucleophilic addition of thiol to C-5 of compound **1a**, which is followed by re-aromatization (Fig. 2a, Supplementary Fig. 2). With C-5 hindered by methyl substitution, 5-methyl-1,4-nitroimidazole **2c** was unreactive with cysteine and remained intact under assay conditions, supporting this reaction mechanism (Supplementary Fig. 3).

The high reactivity of compound **1a** prompted us to investigate its ability to modify cysteine residues in peptides. Tripeptide **2a** was therefore synthesized and treated with compound **1a** in HEPES buffer at pH 7.0. High-performance liquid chromatography (HPLC) analysis showed that 92% conversion of peptide **2a** was achieved within 1 min, and full conversion was achieved after 5 min by generating peptide conjugate **1c** (Fig. 2b, Supplementary Fig. 4). The reaction was significantly accelerated by elevating the pH of the buffer to 8.0, leading to full conversion in only 15 s (Fig. 2b, Supplementary Fig. 5). We next evaluated the pH dependence of this reaction. After incubating 1,4-DNIm **1a** with tripeptide **2a** at pH 7.0–9.0 for 2 min, near-quantitative modification was observed (Fig. 2c, Supplementary Figs. 4-6). Prolonging the incubation time to 30 min led to 28% and 74% conversion of tripeptide **2a** even under acidic conditions at pH 4.0 and 5.0, respectively, demonstrating the wide applicable pH range of this reaction (Fig. 2c, Supplementary Figs. 7–10).

Next, we evaluated the chemoselectivity of compound **1a** under aqueous conditions. Lysines are rich in natural proteins and usually solvent-exposed on protein surface, therefore representing a major competition. In a competitive assay, incubation of tripeptide **2a** (0.1 mM) and a lysine-containing tripeptide **3a** (1.0 mM) with compound **1a** (0.12 mM) in HEPES buffer at pH 7.0 yielded conjugate **1c** near quantitatively (Supplementary Fig. 11). No lysine adduct of tripeptide **3a** was detected by liquid chromatography–mass spectrometry (LC-MS) analysis, demonstrating the dominating selectivity of 1,4-DNIms towards cysteine thiol over lysine ε-amine. Furthermore, compound **1a** did not react with nucleophilic amino acids, including Tyr, Ser, Asp, Arg, Met, and His, under neutral aqueous conditions, further establishing the thiol specificity of 1,4-DNIms (Fig. 2d). Perfluoroaryl compounds have shown great potential in cysteine bioconjugation of peptides[13]. To compare their efficiency with 1,4-DNIms, we chose 6,6′-sulfonylbis(1,2,3,4,5-pentafluorobenzene) (**pf1**), the most reactive perfluoroaryl reagent reported so

**a** Cross-reactivity of Cys and Lys bioconjugation reagents

**b** 1, 4-Dinitroimidazoles react with cysteine and lysine through distinct mechanisms

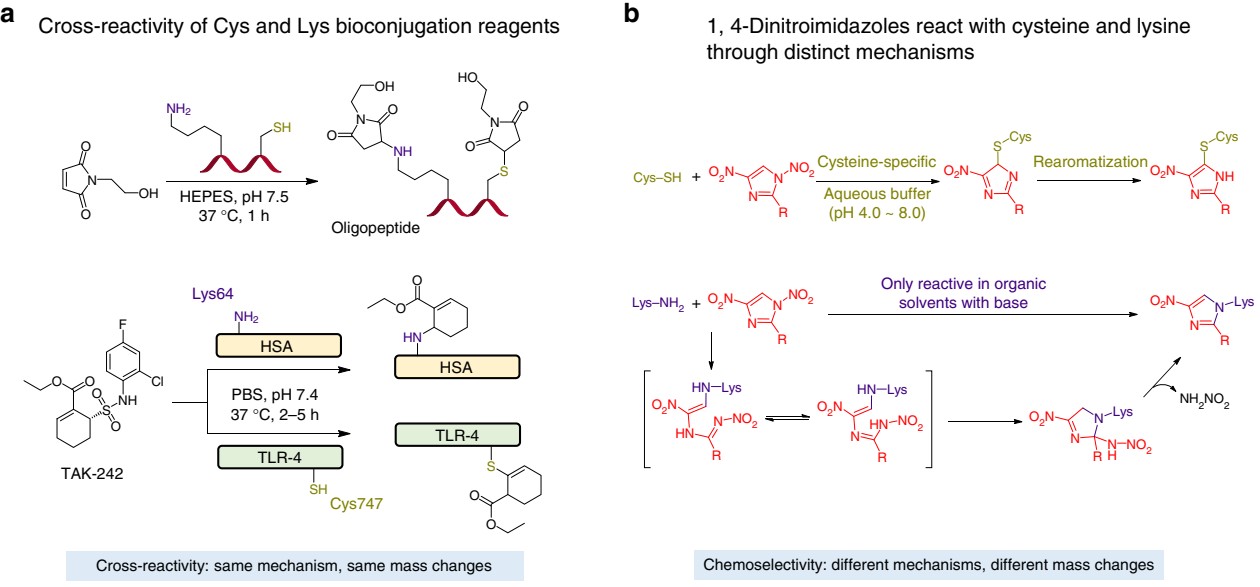

**c** **This work**: highly controllable modification of peptides and proteins by 1, 4-DNIms

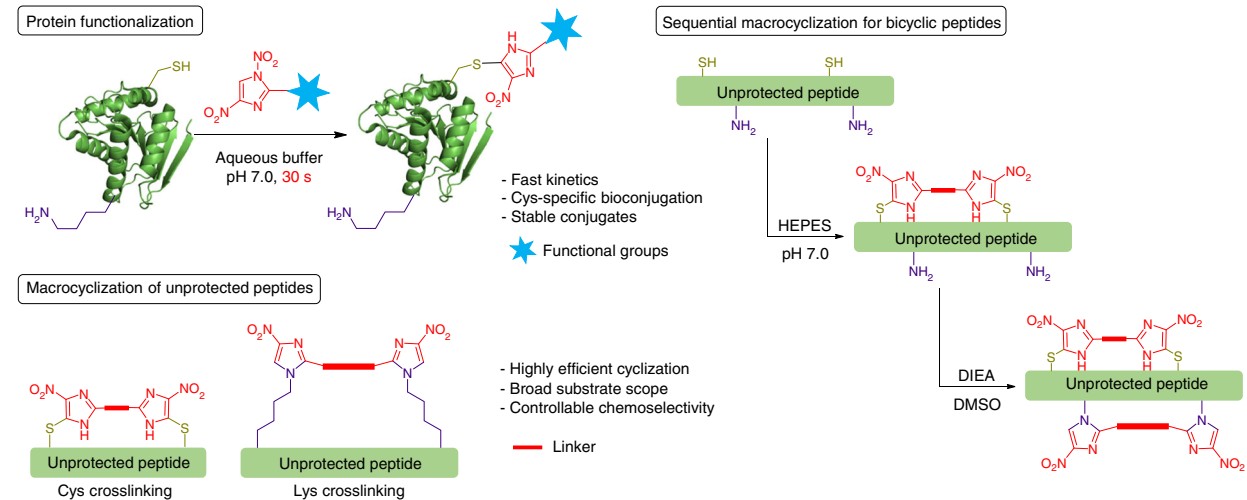

**Fig. 1** 1,4-Dinitroimidazoles as bifunctional bioconjugation reagents. **a** Cross-reactivity of cysteine and lysine modification reagents. **b** 1,4-DNIms react with cysteine and lysine through two distinct mechanisms and display different chemoselectivity under aqueous and organic solvents. **c** Application of 1,4-DNIms in protein functionalization and peptide macrocyclization

far[30], as the model compound. First, we observed that compound **pf1** exhibited a very low solubility (<0.05 mM) in aqueous solutions, whereas 1,4-DNIm **1a** has a solubility higher than 10 mM. In HEPES buffer, pH 7.4, (**pf1**) at a concentration of 0.04 mM was unreactive with peptide **2a**, whereas 1,4-DNIm **1a** modified (**2a**) near quantitatively (Supplementary Fig. 12). This result shows that under neutral aqueous conditions, 1,4-DNIms are superior to perfluoroaryl compounds in terms of solubility and reactivity towards cysteine residues. Together, under acidic to neutral aqueous conditions, 1,4-DNIms are highly efficient and cysteine-specific bioconjugation reagents.

The stability of protein conjugates is a major concern for bioconjugation reagents. Therefore, we evaluated the stability of 1,4-DNIm **1a** and the resulting (4-nitroimidazole)-thiol products. HPLC analysis showed that *N*-phenyl maleimide underwent complete hydrolysis in 2 h in 100 mM phosphate-buffered saline (PBS) buffer at pH 7.4, yielding the maleamic acid (Supplementary Fig. 13). In contrast, compound **1a** displayed resistance to hydrolysis, remaining unchanged for up to 10 h (Supplementary

Fig. 14). Next, we investigated the stability of the (4-nitroimidazole)-thiol linkage. Thiol-maleimide and thiol-arylation conjugates are reported to degrade under acidic, basic, or oxidative conditions[25,30,31]. We incubated conjugate **1c'** in aqueous buffer at pH 2.0, pH 10.0, or in the presence of 10 mM $H_2O_2$ for 10 h at 37 °C. HPLC analysis indicated that conjugate **1c'** did not decompose under tested conditions (Fig. 2e, Supplementary Fig. 15), whereas the *N*-ethyl maleimide conjugate of **2a'** underwent complete hydrolysis at pH 10.0 (Supplementary Fig. 16). Together, both 1,4-DNIm **1a** and its cysteine conjugate exhibited enhanced stability compared with maleimides and maleimide-cysteine conjugates under various conditions, offering an attractive alternative for cysteine bioconjugation of proteins.

Protein conjugates bearing different classes of functionalities, such as fluorophores and biomolecules, are valuable in the fields of biomaterials and chemical biology. To test the applicability of 1,4-DNIms in protein modification and their compatibility with diverse functional components, we synthesized fluorescent 7-nitrobenzo-Z-oxa-1,3 diazole (NBD)-conjugate **4a** and peptide

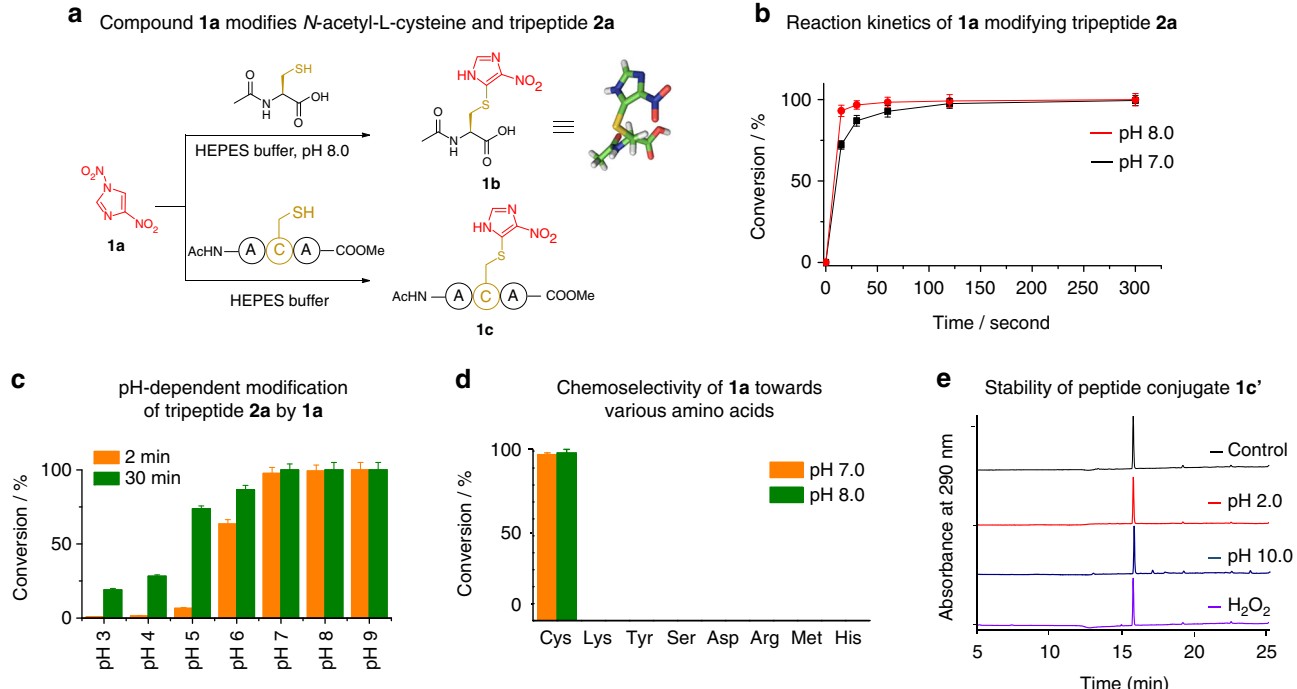

**Fig. 2** Compound **1a** modifies cysteine thiol with high efficiency and specificity by generating stable conjugates. **a** Reaction of compound **1a** with cysteine and tripeptide **2a**. The CCDC reference number of compound **1b** is 1860451. **b** Reaction kinetics of compound **1a** (0.2 mM) with tripeptide **2a** (0.1 mM) at pH 7.0 and 8.0. **c** The reaction between of **1a** (0.2 mM) and **2a** (0.1 mM) under various pH for 2 min and 30 min. **d** Reactions of compound **1a** (0.2 mM) with nucleophilic amino acids (0.1 mM) for 15 min under pH 7.0 and 8.0. **e** Chemical stability of the tripeptide-(4-nitroimidazole) conjugate **1c'** at pH 2.0, pH 10.0, and under oxidative conditions (10 mM $H_2O_2$ in $Na_2CO_3$ buffer at pH 8.0) at 37 °C for 10 h. The conversion values represent means ± SDs (standard deviations) of three independent experiments

arginine-glycine-aspartic acid (RGD)-conjugate **4b** bearing a 1,4-DNIm warhead for cysteine bioconjugation (Fig. 3a). Sortase A (SrtA) with one free cysteine (Cys192) and 26 lysines was selected as a model protein. It was expected that 1,4-DNIms could achieve cysteine-specific labeling by overcoming the challenge of a large number of surface-exposed Lys under neutral aqueous conditions. Incubation of SrtA (20 μM) with compound **4a** (200 μM) for 1 h in PBS buffer, pH 7.0, resulted in quantitative conversion to NBD-modified SrtA, as determined by LC-MS analysis (Fig. 3c). The NBD-modified SrtA was further digested by trypsin and subjected to tandem mass spectrometry (MS) analysis. Results showed that the SrtA$_{186–205}$ segment carried the NBD modification at the Cys192 residue (Supplementary Fig. 17). To further confirm the Cys-specific labeling, SrtA was pre-treated with iodoacetamide (IAA) before incubation with NBD-(1,4-DNIm) **4a**. No fluorescent labeling was observed for the IAA-treated SrtA, further indicating that 1,4-DNIms selectively modified cysteine residues (Fig. 3d). RGD peptides bind to the integrin receptors of cells and are widely used for targeted delivery of chemotherapeutics[32,33]. Compound **4b** with a 1,4-DNIm moiety at the N terminus of RGD sequence was then incubated with SrtA under neutral buffer. To our delight, full conversion to RGD-SrtA conjugate was observed after 1 h (Fig. 3c). These results demonstrate that 1,4-DNIms are compatible with diverse functional components during protein modification.

To make 1,4-DNIms as a general approach to prepare diverse protein conjugates, we synthesized compound **4c** bearing an alkyne group for further functionalization through 1,3 dipolar azide–alkyne cycloaddition (AAC) (Fig. 4a). Bovine serum albumin (BSA), with a single free cysteine (Cys34) and 59 lysine residues, was chosen as a model protein. Treatment of BSA (100 μM) with compound **4c** (500 μM) led to quantitative modification in 1 min, as determined by LC-MS analysis (Fig. 4b). To identify

the modification site, the resulting BSA-**4c** conjugate was reduced, digested by trypsin, and subjected to LC-MS/MS analysis. Results showed that the modification occurs exclusively at the Cys34 residue (Fig. 4d), and no lysine modification is detected. Subsequently, fluorescent labeling of the BSA-**4c** conjugate was realized with a fluorogenic azide, sulfo-Cy3-azide, through CuAAC reaction. Sodium dodecyl sulfate-polyacrylamide gel electrophoresis (SDS-PAGE) analysis showed that BSA was successfully labeled with fluorophore Cy3 through the alkyne handle of compound **4c** as control reactions without copper or Cy3-azide failed to produce any observable fluorescence (Fig. 4c). Therefore, the successful incorporation of an alkyne group as the click reaction handle through 1,4-DNIm bioconjugation provides a facile method for cysteine-selective addition of various payloads to proteins of interest.

The potential toxicity of the thiol-(4-nitroimidazole) linkage generated by bioconjugation is an important concern for the biological application of the 1,4-DNIm chemistry. To address this issue, we synthesized a fluorescein isothiocyanate (FITC)-labeled RGD peptide and an (1,4-DNIm)-modified cyclic RGD (cRGD) peptide (Supplementary Fig. 18). At a concentration of 10 μM, RGD peptides were efficiently uptaken by 293T cells, as demonstrated by the cell staining assay of FITC-RGD. Upon incubation with 10 μM cRGD-(4-nitroimidazole) conjugate, the viability of 293T cells was not affected (Supplementary Fig. 18), indicating that the thiol-(4-nitroimidazole) is not toxic under assay conditions.

**Cys-specific peptide stapling by bisfunctional 1,4-DNIms.** Cyclic peptides have great potential as therapeutic agents and biological tools, and thus the development of stapling reagents is of particular interest[13,18,34–37]. The capability of 1,4-DNIms in

selective cysteine modification prompted us to explore their application in peptide stapling. Thus, a bisfunctional 1,4-DNIm (**5a**) was synthesized and incubated with unprotected peptides containing two cysteine residues at varying sites under neutral aqueous buffer for 10 min. Sodium ascorbate was added to the reaction to avoid disulfide formation caused by air oxidation. High conversions were observed for all peptides examined by yielding corresponding peptide macrocycles, as determined by HPLC analysis (Fig. 4e). In particular, Cys-specific crosslinking of peptide **P3** with unprotected N terminus was achieved, demonstrating the chemoselectivity of 1,4-DNIms towards thiol groups over N-terminal amine.

## 1,4-DNIms modify lysine amine in organic solvents.

Although lysines and N-terminal amine groups in proteins are inert to 1,4-DNIm treatment under neutral aqueous conditions, reactions between amines and 1,4-DNIms have been reported in the synthesis of nitroimidazole derivatives in organic solvents. Early evidence and recent computational studies suggest that the reaction is initiated by nucleophilic attack at C5 position of 1,4-DNIms and follows an ANRORC-like mechanism, consisting of the *a*ddition of a *n*ucleophile followed by *r*ing-*o*pening and *r*ing *c*losure step (Fig. 1b)[38–40]. To provide mechanistic insight of this reaction, we utilized [15]N-labeled aniline to react with 1,4-DNIm (**1a**), generating product **1e** in 82% (Fig. 5a). High-resolution MS (HRMS) analysis showed that (**1e**) is isotopically labeled, indicating that the [15]N atom from aniline is present in the product (Fig. 5a). NMR characterization of (**1e**) by analyzing [15]N-C and [15]N-H coupling signals further indicated that the [15]N of analine was indeed incorporated into the newly formed imidazole ring by forming two C-[15]N bonds with C2 and C5 of (**1a**) (Supplementary Figs. 19–24). Furthermore, X-ray crystallographic analysis unambiguously confirmed the structure of (**1e**) (Fig. 5a).

To examine whether the reaction is initiated by nucleophilic attack at the C5 positions of 1,4-DNIms, we employed 4-(2-aminoethyl)phenol to react with compound **1a**, 5-methyl-1,4-DNIm **2c** and 2-methyl-1,4-DNIm **2d** (Supplementary Fig. 25). Results showed that compounds **1a** and **2d** reacted with similar efficiency, leading to 52% and 50% conversion, respectively. In contrast, 5-methyl-1,4-DNIm **2c** was inert under assay conditions (2% conversion). No accumulation of ring-opening intermediates was observed in these reactions. Such substitution effect supports C5 position as the site for nucleophilic addition by amines.

To examine the proposed ring-opening step, we chose 2-isopropylaninline and 2-methyl-1,4-DNIm (**2d**) as reacting partners (Fig. 5b), in which the bulky substitutions are expected to slow down the ring closure and facilitate the capture of ring-opening intermediates. To our delight, in addition to the final product **1f**, the accumulation of a labile intermediate **1g** was observed by thin layer chromatography (Supplementary Fig. 26). HRMS analysis of the reaction mixture also revealed an intermediate with a mass matching (**1g**) (Fig. 5c). The ultraviolet–visible spectra of product **1g** display a characteristic absorbance at 369 nm (Fig. 5c), suggesting the presence of an extended π-conjugation resulted from the opening of imidazole ring. Furthermore, gas chromatography-MS analysis of intermediate (**1g**) yielded molecular fragments that match the proposed structure of (**1g**) (Supplementary Fig. 27). Although we were not able to perform NMR analysis of (**1g**) due to its spontaneous conversion to product **1f**, our data support the proposal that 1,4-DNIms react with amines through the ANRORC-like mechanism.

Next, we explored the application of 1,4-DNIms in Lys bioconjugation of peptides. After screening various reaction

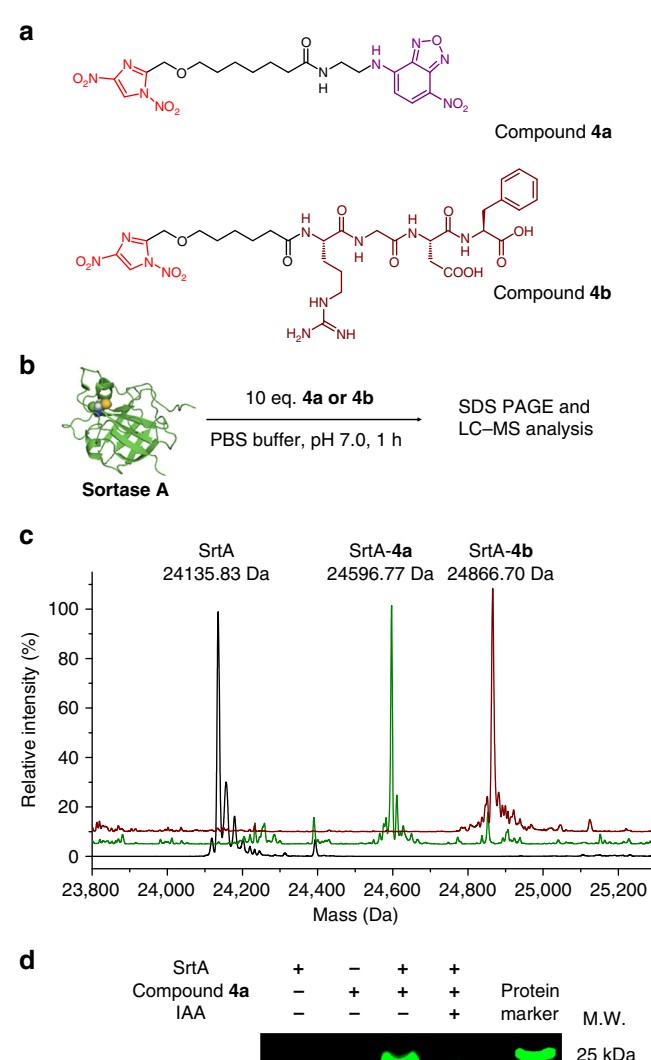

**Fig. 3** Modification of SrtA with 1,4-DNIms bearing a fluorescent NBD moiety and an RGD peptide. **a** Chemical structures of 1,4-DNIm derivatives **4a** and **4b**. **b** Bioconjugation of SrtA by 1,4-DNIm derivatives. Conditions: SrtA (20 μM), compound **4a** or **4b** (200 μM), 100 mM PBS buffer, pH 7.0, 1 h. **c** LC-MS analysis of SrtA modified by compounds **4a** and **4b**. Molecular mass (average): SrtA calcd. 24,136.03 Da, observed 24,135.83 Da; (SrtA-**4a**) conjugate calcd. 24,596.44 Da, observed 24,596.77 Da; (SrtA-**4b**) conjugate calcd. 24,866.77 Da, observed 24,866.70 Da. **d** SDS-PAGE analysis of the fluorescent labeling of SrtA or iodoacetamide-treated SrtA by compound **4a**

conditions, we found that in DMSO with DIEA (*N*,*N*-diisopropylethylamine) added as the base, compound **1a** (0.1 mM) was able to modify a lysine-containing tripeptide **3a** (0.1 mM) with near-quantitative conversion in 30 min (Fig. 6a). HRMS analysis and NMR analysis determined that product **1d** is a ε-(4-nitroimidazole)-Lys derivative (Fig. 6a, Supplementary Figs. 28–32). Next, we explored the chemoselectivity of compound **1a** in DMSO via incubation with nucleophilic amino acids, including Tyr, Ser, Trp, His, Asp, and Arg. No reaction occurred with these amino acids, indicating that this reaction was lysine-specific. In comparison with perfluoroaryl reagent **pf1**, 1,4-DNIm **1a** displayed similar reactivity when modifying a lysine-containing tripeptide in DMSO (Supplementary Fig. 33). The 4-nitroimidazole linkage resulted

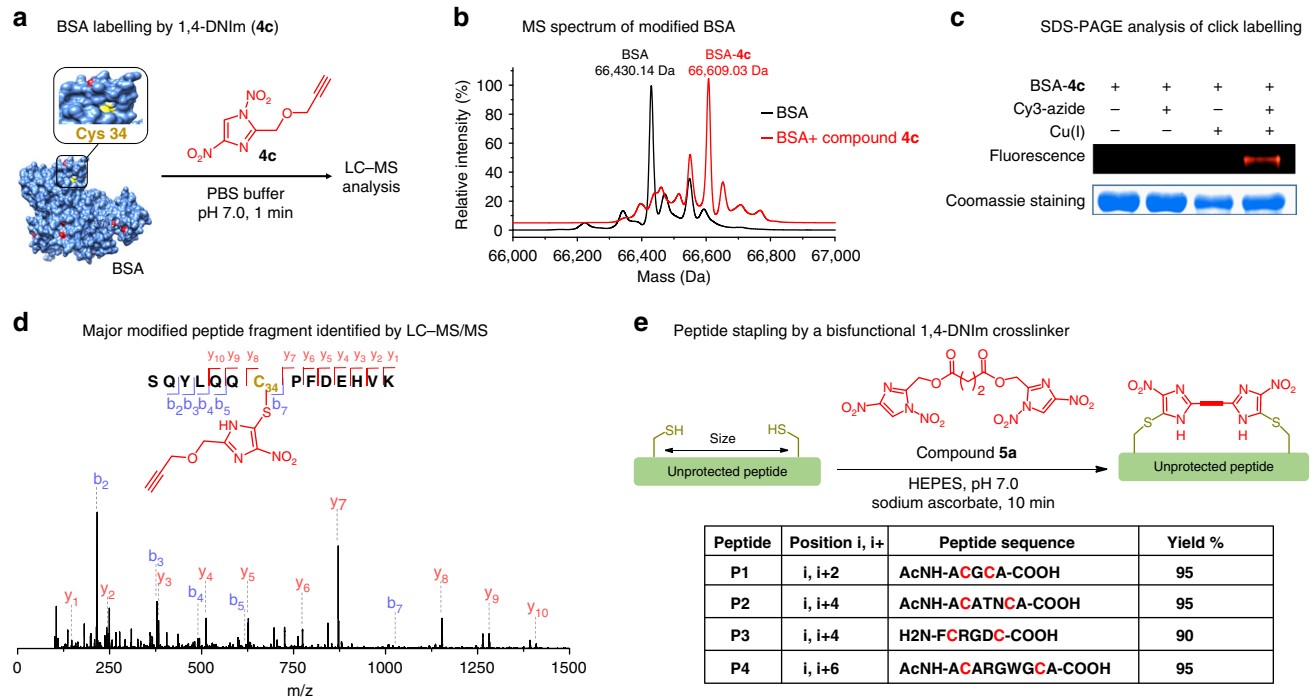

**Fig. 4** Thiol-specific labeling of BSA protein and peptide stapling by 1,4-DNIm derivatives. **a** Bioconjugation of derivative **4c** with BSA. The free thiol of Cys34 residue is highlighted in yellow. **b** ESI-MS spectra of BSA protein before and after treatment with compound **4c**. Molecular mass (average): BSA calcd. 66,429.98 Da, observed 66,430.14 Da; BSA-**4c** calcd. 66,609.12 Da, observed 66,609.03 Da. **c** Gel image of the reaction between BSA-**4c** conjugate and sulfo-Cy3-azide by a CuAAC reaction. **d** LC-MS/MS analysis of a modified peptide fragment from trypsin digestion of BSA-**4c** conjugate. **e** Macrocyclization of cysteine-containing unprotected peptides by a bisfunctional 1,4-DNIm (**5a**). Conditions: peptides (1.0 mM) and compound **5a** (1.0 mM) and sodium ascorbate (2.0 mM) were incubated in 100 mM HEPES buffer at pH 7.0 for 10 min before acidified with formic acid and subjected to HPLC analysis

from Lys-(1,4-DNIm) bioconjugation was highly stable and remained intact after incubation under acidic (pH 2.0), basic (pH 10.0), and oxidative (10 mM $H_2O_2$) conditions at 37 °C for 10 h (Supplementary Fig. 34). To sum up, 1,4-DNIms react specifically with cysteine thiols under neutral aqueous conditions, and modify lysine $\varepsilon$-amines through a distinct mechanism with high selectivity in organic solvents in the absence of thiols by generating stable conjugates.

**Lys-specific peptide stapling by bisfunctional 1,4-DNIms**. Next, we applied bisfunctional 1,4-DNIm **5a** in peptide macrocyclization by crosslinking two lysine residues. By varying the site of lysine residues ($i,i + 4$ to $i,i + 12$), a series of peptide macrocycles were prepared in high yields (Fig. 6b). Diaminopropionic acid (Dap) in peptide **P7-Dap** was also efficiently crosslinked, demonstrating the versatility of this protocol. Peptide macrocyclization usually stabilizes the secondary structures of peptides and improves their proteolytic stability. Indeed, circular dichroism analysis showed that cyclization of peptide **P8** increased the content of α-helical structure (Supplementary Fig. 35). Regarding the proteolytic stability of peptides, linear peptide **P7** was completely hydrolyzed in the presence of proteinase K in 30 min, whereas cyclized **P7** was significantly more stable and remained intact for up to 4 h (Supplementary Fig. 36).

**Preparation of bicyclic peptides by bisfunctional 1,4-DNIms**. The synthesis of peptide architectures with overlapping rings is particularly challenging because it often requires appropriate orthogonal ring-closing reactions, as well as the incorporation of functionalized unnatural amino acids[41,42]. The controllable chemoselectivity of 1,4-DNIms under aqueous and organic solvents

provides a convenient method to integrate two consecutive stapling reactions and to enable regioselective construction of complex bicyclic peptides. For this purpose, we synthesized a model peptide **P10** containing two cysteine and two lysine residues for $i,i + 5$-position macrocyclization (Fig. 6c). Incubation of stapling reagent **5a** (0.1 mM) with peptide **P10** (0.1 mM) in 100 mM HEPES buffer at pH 7.0 resulted in quantitative conversion to the cyclic product **P11** in 30 s as determined by LC-MS (Supplementary Fig. 37). The Cys-specific crosslinking are confirmed by the lack of fragmentation between two cysteine residues during tandem MS analysis (Fig. 6c, Supplementary Fig. 38). The reaction mixture of **5a** and **P10** was then lyophilized without purification, redissolved in DMSO, and treated with another 2 equiv compound **5a**. Results showed that a second crosslinking between lysine residues in the monocyclic peptide **P11** was achieved in 30 min by yielding a bicyclic peptide **P12** with overlapping ring structures, as indicated by tandem MS analysis (Fig. 6c, Supplementary Fig. 39). Thus, bisfunctional 1,4-DNIms are powerful peptide stapling reagents that allow controllable consecutive peptide stapling to generate peptide scaffolds with entangled macrocycles.

## Discussion

In summary, we have demonstrated that 1,4-DNIms are highly efficient, bifunctional bioconjugation reagents that provide stable protein conjugates and peptide macrocycles. This chemistry allows facile incorporation of various functional groups into proteins at defined positions. Importantly, 1,4-DNIms represent a class of electrophilic bioconjugation reagents that react with cysteine and lysine through completely different mechanisms. Consequently, 1,4-DNIms display vastly different reactivity and chemoselectivity under different reaction conditions, making

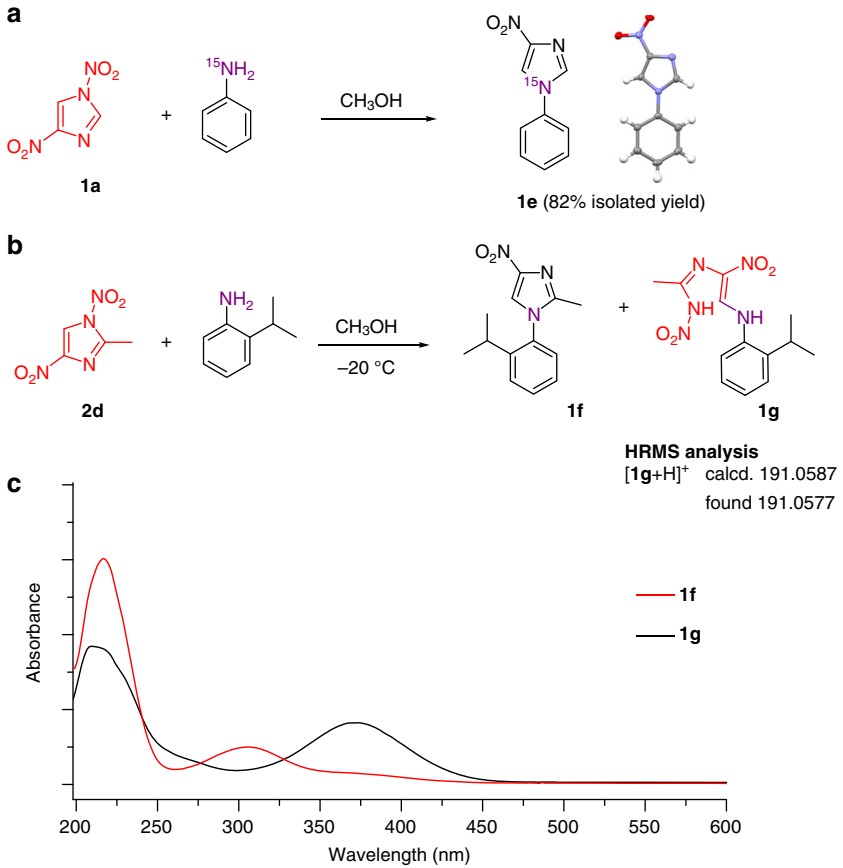

**Fig. 5** Mechanistic investigation of the reactions between 1,4-DNIms and amines. **a** 1,4-DNIm **1a** reacts with [15]N-labeled aniline to produce (**1e**). The CCDC reference number of compound **1e** is 1876959. HRMS (ESI) analysis of product (**1e**): $m/z$ calcd. for $C_9H_8N_2{}^{15}NO_2$ [M + H]$^+$ 191.0587, found 191.0577; $m/z$ calcd. for $C_9H_7N_2{}^{15}NNaO_2$ [M + Na]$^+$ 213.0406, found 213.0400. **b** 2-isopropylaninline reacts with 2-methyl-1,4-DNIm **2d**. HRMS (ESI) analysis of intermediate (**1g**): $m/z$ calcd. for $C_{13}H_{17}N_5NaO_4$ [M + Na]$^+$ 330.1178, found 330.1172; **c** UV–Vis characterization of products **1f** and **1g**

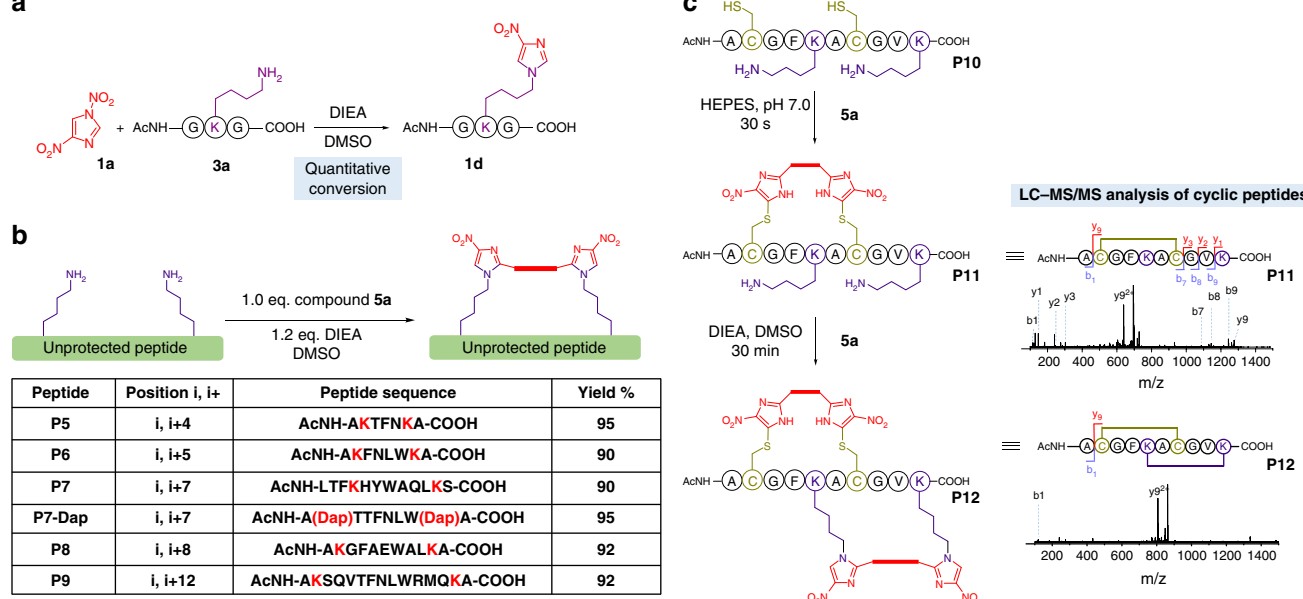

**Fig. 6** Macrocyclization of peptides by a bisfunctional 1,4-DNIm through lysine and cysteine crosslinking. **a** Lysine modification by 1,4-DNIm **1a**. **b** Peptide macrocyclization by bisfunctional 1,4-DNIm **5a**. **c** Preparation of a bicyclic peptide by controllable sequential macrocyclization by 1,4-DNIm **5a**

| Peptide | Position i, i+ | Peptide sequence | Yield % |
|---|---|---|---|
| P5 | i, i+4 | AcNH-A**K**TFN**K**A-COOH | 95 |
| P6 | i, i+5 | AcNH-A**K**FNLW**K**A-COOH | 90 |
| P7 | i, i+7 | AcNH-LTF**K**HYWAQL**K**S-COOH | 90 |
| P7-Dap | i, i+7 | AcNH-A**(Dap)**TTFNLW**(Dap)**A-COOH | 95 |
| P8 | i, i+8 | AcNH-A**K**GFAEWAL**K**A-COOH | 92 |
| P9 | i, i+12 | AcNH-A**K**SQVTFNLWRMQ**K**A-COOH | 92 |

them highly controllable in protein modification without cross-reactivity. In view of the kinetics, selectivity, and stability of the 1,4-DNIm bioconjugation reactions, 1,4-DNIm compounds are a valuable addition to the current toolbox for peptide and protein bioconjugation.

## Methods

**General information**. All the reagents and solvents were obtained from Sigma-Aldrich, Alfa-Aesar, or Acros, and used directly without further purification. Amino acids and derivatives were obtained from commercial sources. BSA was purchased from BBI Life Sciences and used without further purification. SrtA 5M was a kind gift from Prof. Yi Cao at the Nanjing University. NMR spectra were recorded on Bruker AMX-400 instrument for $^1$H NMR at 400 MHz and $^{13}$C NMR at 100 MHz, using tetramethylsilane as an internal standard. The following abbreviations (or combinations thereof) were used to explain multiplicities: s = singlet, d = doublet, t = triplet, q = quartet, m = multiplet, and br = broad. Coupling constants, $J$, are reported in Hertz units (Hz). HRMS were recorded on an Agilent Mass spectrometer using ESI-TOF (electrospray ionization-time of flight). HPLC profiles were obtained on Agilent 1260 HPLC system using commercially available columns.

**Modification of peptides and proteins by 1,4-DNIms**. Typically, peptide (0.1 mM) was incubated with 1,4-DNIm (0.1 mM) and in HEPES buffer (100 mM) at indicated pH at 25 °C for indicated time before analyzis by HPLC and MS. For protein modification, 20 μM protein was incubated with 10 eq. of 1,4-DNIm derivative in PBS buffer at pH 7.0 at 25 °C for 1 h. The reaction mixture was quenched by the addition of 1% HCOOH or removing the excess compound by PD10 desalting column before subjecting to LC-MS analysis.

**Fluorescent labeling of BSA via CuAAC reaction**. BSA-**4c** conjugate was prepared by incubating 15 μM BSA and 5 eq. compound **4c** in 100 mM PBS buffer at pH 7.0. The excess compound **4c** was then removed by PD10 desalting column, and the BSA-**4c** conjugate was eluted with 100 mM PBS buffer, pH 7.2. BSA-**4c** conjugate (15 μM) was further incubated with 100 μM sulfo-Cy3-azide, 250 μM CuSO$_4$, 500 μM BTTAA (bis[(tertbutyltriazoyl)methyl]-[(2-carboxymethyltriazoyl) methyl]amine), and 2.5 mM freshly prepared sodium ascorbate for 1 h at 25 °C before quenching with 5 mM BCA. The reaction samples were analyzed by SDS-PAGE gels and imaged by Gel Doc$^{TM}$ XR+ (Bio-Rad). The protein gels were also stained by Coomassie Brilliant Blue.

## Data availability

The X-ray crystallographic coordinates for structures of compounds **1b** and **1e** have been deposited at the Cambridge Crystallographic Data Center (CCDC), under deposition numbers of 1860451 and 1876959, respectively. All relevant data are available in supplementary information and from the authors.

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

## Acknowledgements

This work is supported by the 1000-Youth Talents Plan, NSF of China (Grant 21778030), NSF of Jiangsu Province (Grant BK20160640), and the Fundamental Research Funds for the Central Universities (Grants 14380138 and 14380131). We thank Prof. Yi Cao (Nanjing University) for the gift of SrtA protein and Prof. Qi Zhang (Fudan University) for helpful discussion during the preparation of the manuscript.

## Author contributions

Q.L. designed and carried out the chemical synthesis and reactions. Y.T. performed most of the biochemical experiments regarding protein modifications. W.S. performed the cell culture, staining experiments, and the cellular toxicity assay. L.J. supported biochemical experiments. H.W. designed and supervised the project. All authors discussed the results and commented on the manuscript. Q.L., Y.T., and H.W. wrote the manuscript. All authors have given approval to the final version of the manuscript.

## Additional information

**Competing interests:** All authors declare no competing interests.

