## [Peer Review File · Nature Communications]

Reviewers' comments:

Reviewer #1 (Remarks to the Author):

The manuscript of Wang and coworkers entitled "Dinitroimidazoles as bifunctional bioconjugation reagents for Protein Functionalization and Peptide Macrocyclization" describes a novel reagent for the modification of proteins and peptides. The authors demonstrate that by carefully selecting the reaction conditions, either cysteine or lysine residues can be modified. The dinitroimidazole reagents selectively modify cysteine residues under near neutral, aqueous conditions, but under alkaline conditions in organic solvents also lysine residues are modified. The authors exploit this reactivity elegantly for the preparation of a bicyclic peptide and in my opinion this is the most important selling point of the strategy. For example, the chemistry would allow the preparation of bicyclic lantibiotic analogues that cannot be prepared with traditional ligation strategies.

Another advantage of the reagents compared to other strategies is the exquisite cysteine selectivity (on a protein level) and the fast reaction. According to the authors, this should make the reagents valuable tools for the preparation of bioconjugates and for proteomics purposes. The selectivity and reactivity have thus far been shown on recombinant purified proteins and this is sufficient for the preparation of bioconjugates. However, for the application of the reagents in proteomics purposes, the selectivity and reactivity should be assessed on a far large number of proteins and on more complex samples. Many proteins, in particular enzymes have so-called "hyperreactive" cysteine and lysine residues that may react with the reagent as well. In my opinion, assessing the labeling on two proteins and several peptides therefore is not sufficient to support the claim that the reagent is suitable for proteomics purposes. Neither does it support the claim that it has exquisite selectivity. To support such claims, a global profiling study on complex samples (for example a lysate) should be performed. And the modified residues should be carefully analyzed.

The final advantage of the reagents is the stability of the formed adduct. The products reveal to be stable under alkaline, acidic and oxidizing conditions. Even though I agree with the authors that this is indeed an advantage over the well-established maleimide chemistry, I do think that the authors overlook the major advances that have been made to stabilize the maleimide addition products. For example the adducts have been stabilized by promoting hydrolysis (Senter, P. D. and coworkers (Nature biotechnology, 2014, 32(10), 1059-1062) and or by exploiting exo-cyclic Michael acceptors (Parthasarathy, M. and coworkers Angewandte Chemie 2015, 128 (4), 1454). Furthermore, the toxicity and immunogenicity of the formed maleimide adducts have been studied in detail. For the new nitroimidazole linkages, it is not known if they are toxic/immunogenic. However, to be suitable for the preparation of medically relevant bioconjugates, this is a major point that needs to be addressed. The use of the strategy will be limited to fundamental studies if the linkage proves to be toxic and/or immunogenic.

Overall, I do think that the manuscript is interesting and well-carried out, but current form it is not yet publishable in Nature Communications. However, it could be suitable if the authors address these issues and the other issues raised below

Other points that need to be addressed:

Page 1: "cysteines are rare in human proteins (1.9% of residues) and often exist as disulfides, therefore" should be "cysteines are rare in human proteins (1.9% of residues) and often exist as disulfides, and therefore"

Page 1: "Lysine residues, on the other hand, occurs much more commonly in proteins" should be "Lysine residues, on the other hand, occur much more commonly in proteins"

Page 2: In the text, the authors state that the mechanism is depicted in Figure 1B. I do not think that this is a proper mechanism. It only gives the site of attack. The mechanism provided in the SI

is more accurate, but lacks the rearomatization step.

Page 2: The authors state "With C-5 hindered by methyl substitution, 5-methyl-1,4-nitroimidazole 2c was unreactive with cysteine under assay conditions, supporting this reaction mechanism (Fig.S2)." I do not think that this single result is sufficient to conclude this. The second step of the reaction is re-aromatization, which is not feasible with the methyl substituted derivative. The degradation products could provide insight in the reaction mechanism and side reactions.

Page 2 and 3: The authors describe the results of the peptide labeling reactions. The efficiency has been determined by HPLC. Based on the traces, it is difficult to determine this, since the peak of DNIM and the non-modified peptide co-elute in many of the chromatograms. Furthermore, the reaction results in several degradation products of dinitroimidazole 1a. Some of these products peaks are also present in the control (conjugate 1c). The referee cannot conclude based on the presented data if these products indeed resulted from 1a or that they are degradation products of conjugate 1c. However, if these peaks are indeed caused by degradation products from 1a, the results do not match with the outcome of the stability studies. The authors show that DNIM 1a is stable in HEPES. Since degradation products of 1a are formed during the conjugation reaction, it is tempting to speculate that these degradation products are formed due to the presence of thiol/peptide. This could indicate a side-reaction and it is important to describe in more detail what these compounds are (see also the other points before).

Page 3: The authors write "Incubation of SrtA (20 μ M) with compound 4a (200 μ M) for 1 h in PBS buffer, pH 7.0 resulted in quantitative conversion to NBD-modified SrtA, as determined by LC-MS analysis (Fig. 3C)." Based on the provided data it is difficult to determine the conversion. It would be better if the chromatogram would be given with the peaks of the modified and unmodified protein, rather than only the deconvoluted MS spectrum of the modified protein. The MS also does not provide information on the modification site and it can therefore not be concluded that the reagent modifies the active site cysteine of sortase (even though I agree this is very likely). The authors do identify the modification site correctly for BSA.

Page 3: Even though the gel-labeling results for sortase (Fig 3) are convincing, the authors should perform a competition experiment with iodoacetamide to demonstrate that the reagent indeed binds to the cys residue. Pretreating the protein with this reagent should block labeling if the reaction is indeed selective for cys. (This is a biochemical alternative for identification of the modification site by MS).

Page 4: Figure 4B: It is difficult to see the mass shift in the MS spectrum. It would be better if a different zoom would be provided.

Page 4: For the stapling of the cys peptides, the authors used ascorbate. I presume this is to prevent disulfide formation, but it is not clearly stated in the text.

Page 4: The authors conclude "To sum up, under different conditions, 1,4-DNIMs selectively react with cysteine thiols or lysine ϵ -amines through two distinct mechanisms, which may account for their vastly different reactivity and chemoselectivity under aqueous and organic conditions." I do not think this conclusion is fully supported. For example, the authors do not evaluate the reactivity of cysteine residues under organic conditions. Thiols may also react under these conditions. In my opinion, with the here presented data it can therefore not be claimed that 1,4-DNIMs react selectively with lysine residues in organic solvent. Either the experiment with a peptide containing a thiol and an amine should be added or the authors should claim that the 1,4-DNIMs react selectively with lysines in organic solvents in the absence of thiols.

Supporting information:

The $^1\text{H-NMR}$ of 1b has too many protons. I cannot assign the double triplet at 3.27 ppm. Peaks

are lacking in the carbon spectrum.

For compound m3, it is not clear how the regioselectivity is determined. Even though this is the most likely regioisomer, it should still be shown. Furthermore, the HRMS is difficult to read due to the low quality. Finally, it is not clear what is depicted in the top and the bottom spectrum.

Scheme S5: yield of step a) does not match the experimental yield.

For compound m9: the splitting pattern at 1.35 ppm is strange. I do not expect ddd's for this compound. There are no diastereotopic protons and at most a pentet or double triplet is to be expected for the CH₂ at the middle of the linker.

For compound m11: the splitting pattern for the CH₃ of the ester is strange. This should be a triplet.

For compound m13: the splitting pattern at 1.39 ppm is strange. I do not expect ddd's for this compound. At most a pentet or double triplet is to be expected for the CH₂ at the middle of the linker.

Finally, the authors do not state the previously literature on 1,4-dinitroimidazole compounds. These reagents have been used quite often for the modification of amines in organic solvents.

Reviewer #2 (Remarks to the Author):

I have reviewed the paper and there are items of substance that prevent me from offering enthusiastic support. I feel that the authors must include proposed mechanism in the main text and be in a position to support their claims by robust NMR analysis of regioselectivity in each case. I refer, of course, to the nucleophilic substitution at the imidazole ring. Without robust assignment in each of the cases (whether it is lysine or cysteine nucleophile), it is not possible to make a judgment. The authors do offer some mechanistic schemes in the SI but this is definitely not enough. They need to appear in the main section and have robust support. You just cannot take some obscure chemistry from a Polish journal (which cannot be easily found on the web) and expect that people will simply trust that everything is super clear. No. This is not click chemistry, which was extremely well understood prior to its ultimate application in biology. We are dealing with some fairly tricky (but interesting) reaction sequence in the present case and you just cannot dump this stuff on the readership of Nature Communications and expect everyone to trust you. Please make sure all of the compounds are well characterized.

Reviewer #3 (Remarks to the Author):

This manuscript reports a new method of peptide and protein modification of cysteine and lysine sites. The choice of electrophile for this work is novel as it utilizes a 1,4-dinitroimidazole electrophile, which is, to the best of my knowledge without precedent. The work however needs to be put in more context than is currently presented as perfluoroaryl electrophiles (developed by Pentelute lab) can be used in a similar fashion. The advantage of the method described in this manuscript is smaller and could potentially offer advantages over perfluoroaryl electrophiles. In order to provide the readership with greater insight, I would suggest the following information be incorporated into a revised manuscript:

1. Compare and contrast the relative advantages and disadvantages of the 1,4-dinitroimidazole reagent versus perfluoroaryl on a model peptide and a protein. This could be conducted by determining the rate constant of the 1,4-dinitroimidazole versus perfluoroaryl versus maleimide for

example.

2. Degradation products are observed in the model reaction between 1a and 2a (Fig. 2, Fig. S3). Please identify the degradation products as they appear to be formed in reasonable quantities (e.g., 60s, 120s and 300 s HPLC traces in Fig. 3).

3. The authors mention that the product of maleimide couplings (i.e., thiosuccinimide linkage) is prone to hydrolysis. The authors fail to report the recent works of Caddick, Baker and Chudamsama who have developed hydrolytically stable maleimide and pyridazinedione linkages (Bioconjugate Chem 2018, 29, 486; Chem. Sci. 2016, 7, 799). This work needs to be contextualised against this previous research.

4. The presentation of the reaction schemes is poor. Increase the size of the structures and font in the figures.

With the issues addressed, I would consider this work suitable for publication in Nature Commun.

5. Fix geometry of alkyne substituents.

Response to Referees Letter

First, we would like to thank all the reviewers for providing suggestion and comments, which have greatly helped us improve our manuscript. To address the concerns from the reviewers, we have made substantial revision to the original manuscript, including:

1. The recent development of maleimide chemistry for protein bioconjugation has been included in the revised manuscript, as requested by Reviewer #1 and #3;
2. We provide experimental data regarding the toxicity of 4-nitroimidazole linkage generated by 1,4-DNIm bioconjugation, as requested by Reviewer #1;
3. We provide updated HPLC data of reactions between 1,4-DNIm and cysteine-containing peptides, and explain the issue of “degradation products”, as requested by Reviewer #1 and #3;
4. We provide mechanistic investigation of the reactions between 1,4-DNIm and amines, as requested by Reviewer #2.

In addition to these major revisions, we have carefully addressed all the comments from the reviewers, provided missing/additional data to better support our conclusions. Changes in the main text are in red.

A point-by-point response to reviewers' comments are listed below:

Point-to-point response:

In response to Reviewer #1 (quotes from reviewer are in italicized) (Key questions have been marked in red):

The manuscript of Wang and coworkers entitled “Dinitroimidazoles as bifunctional bioconjugation reagents for Protein Functionalization and Peptide Macrocyclization” describes a novel reagent for the modification of proteins and peptides. The authors demonstrate that by carefully selecting the reaction conditions, either cysteine or lysine residues can be modified. The dinitroimidazole reagents selectively modify cysteine residues under near neutral, aqueous conditions, but under alkaline conditions in organic solvents also lysine residues are modified. The authors exploit this reactivity elegantly for the preparation of a bicyclic peptide and in my opinion this is the most important selling point of the strategy. For example, the chemistry would allow the preparation of bicyclic lantibiotic analogues that cannot be prepared with traditional ligation strategies.

*Another advantage of the reagents compared to other strategies is the exquisite cysteine selectivity (on a protein level) and the fast reaction. According to the authors, this should make the reagents valuable tools for the preparation of bioconjugates and for proteomics purposes. The selectivity and reactivity have thus far been shown on recombinant purified proteins and this is sufficient for the preparation of bioconjugates. However, for the application of the reagents in proteomics purposes, the selectivity and reactivity should be assessed on a far large number of proteins and on more complex samples. Many proteins, in particular enzymes have so-called “hyperreactive” cysteine and lysine residues that may react with the reagent as well. **In my opinion, assessing the labeling on two proteins and several peptides***

therefore is not sufficient to support the claim that the reagent is suitable for proteomics purposes. Neither does it support the claim that it has exquisite selectivity. To support such claims, a global profiling study on complex samples (for example a lysate) should be performed. And the modified residues should be carefully analyzed.

Response to the comment:

Thanks for the comment.

We agree with the reviewer that current data mainly demonstrates the applicability of 1,4-DNIm in the preparation of protein conjugates, and we have not provide data to demonstrate their behavior in a complex protein sample. Therefore, we have withdrawn the statements regarding the potential application in proteomics from the manuscript and make this report focus on the chemistry of 1,4-DNIm and their application in the preparation of protein conjugates.

Our initial proposal is that since the modification of Lys and Cys by 1,4-DNIm result in different mass changes, this might be an advantage compared to other protein modification reagents for proteomics that lead to the same mass change when cross-reactivity occurs. Due to the lack of facility and expertise, we are not able to independently perform 1,4-DNIm-based proteomic studies at this moment; however, we are actively involved in a collaboration in this regard. Herein, we would like to share some preliminary data to show that a 1,4-DNIm probe can fish out different proteins from cell lysate under various pH. As shown below, under acidic aqueous conditions, the 1,4-DNIm probe enriched additional proteins from the cell lysate compared to neutral conditions. We are working on this project and hopefully, report the results in the near future.

A) Using a 1,4-DNIm-biotin conjugate to probe reactive cysteine species in cell lysate

B) SDS PAGE and silver stain analysis of the resulting sample

The final advantage of the reagents is the stability of the formed adduct. The products reveal to be stable under alkaline, acidic and oxidizing conditions. Even though I agree with the authors that this is indeed an advantage over the well-established maleimide chemistry, I do think that the authors overlook the major advances that have been made to stabilize the maleimide addition products. For example the adducts have been stabilized by promoting hydrolysis (Senter, P. D. and coworkers (Nature biotechnology, 2014, 32(10), 1059-1062) and or by exploiting exo-cyclic Michael acceptors (Parthasarathy, M. and coworkers

Angewandte Chemie 2015, 128 (4), 1454). Furthermore, the toxicity and immunogenicity of the formed maleimide adducts have been studied in detail. **For the new nitroimidazole linkages, it is not known if they are toxic/immunogenic.** However, to be suitable for the preparation of medically relevant bioconjugates, this is a major point that needs to be addressed. **The use of the strategy will be limited to fundamental studies if the linkage proves to be toxic and/or immunogenic.**

Response to the comment:

We agree with the reviewer that recent development of maleimide chemistry was not properly included in the previous manuscript. Therefore, we have added related information in this revised manuscript. Please see Page 1 (right column, last paragraph), Reference 24-29 for details.

In addition, the reviewer raised an important question regarding the possible toxicity of the nitroimidazole linkage. To address this concern, we synthesized RGD peptide derivatives that contain Cys-(4-nitroimidazole) linkage and Lys-(4-nitroimidazole) linkage, and examined their toxicity towards 293T cell line. Results showed that at 10 μ M concentration, these RGD peptide conjugates enters cell efficiently, but did not exhibit cellular toxicity. Please see these data in Page 3, left column and Supplementary Figure 18.

At this point, we are not able to test the immunogenic effect of nitroimidazole linkages due to the lack of expertise and facility in our lab. But we are actively looking for collaboration to address this issue, and hopefully will provide information in the future.

Other points that need to be addressed:

Page 1: “cysteines are rare in human proteins (1.9% of residues) and often exist as disulfides, therefore” should be “cysteines are rare in human proteins (1.9% of residues) and often exist as disulfides, and therefore”

Response to the comment:

The mistake has been corrected (please see Page 1, first paragraph).

Page 1: “Lysine residues, on the other hand, occurs much more commonly in proteins” should be “Lysine residues, on the other hand, occur much more commonly in proteins”

Response to the comment:

The mistake has been corrected (please see Page 1, first paragraph).

Page 2: In the text, the authors state that the mechanism is depicted in Figure 1B. I do not think that this is a proper mechanism. It only gives the site of attack. The mechanism provided in the SI is more accurate, but lacks the re-aromatization step.

Response to the comment:

Thanks for the comment.

We have provided a detailed reaction mechanism in Fig. 1B and Supplementary Figure 2, and changed the main text accordingly (Page 2, Left column).

Page 2: The authors state “With C-5 hindered by methyl substitution, 5-methyl-1,4-nitroimidazole 2c was unreactive with cysteine under assay conditions, supporting this reaction mechanism (Fig.S2).” I do not think that this single result is sufficient to conclude this. The second step of the reaction is re-aromatization, which is not feasible with the methyl substituted derivative. The degradation products could provide insight in the reaction mechanism and side reactions.

Response to the comment:

Thanks for the comment.

To address this question, we have provided additional HPLC analysis data to show that when 5-methyl-1,4-dinitroimidazole **2c** was incubated with cysteine, no reaction occurred at all. Compound **2c** remained intact after 1 hour incubation and no accumulation of 5-methyl-4-nitroimidazole was observed. This result indicates that the methyl substitution prevents the first step of Cys bioconjugation, which is the nucleophilic attack of thiol to C5 position of 1,4-dinitroimidazole. Please see Supplementary Figure 3.

Page 2 and 3: The authors describe the results of the peptide labeling reactions. The efficiency has been determined by HPLC. Based on the traces, it is difficult to determine this, since the peak of DNIM and the non-modified peptide co-elute in many of the chromatograms. Furthermore, the reaction results in several degradation products of dinitroimidazole 1a. Some of these products peaks are also present in the control (conjugate 1c). The referee cannot conclude based on the presented data if these products indeed resulted from 1a or that they are degradation products of conjugate 1c. However, if these peaks are indeed caused by degradation products from 1a, the results do not match with the outcome of the stability studies. The authors show that DNIM 1a is stable in HEPES. Since degradation products of 1a are formed during the conjugation reaction, it is tempting to speculate that these degradation products are formed due to the presence of thiol/peptide. This could indicate a side-reaction and it is important to describe in more detail what these compounds are (see also the other points before).

Response to the comment:

Thanks for careful examination of HPLC data. We agree with the reviewer that in the original manuscript, 1,4-DNIm **1a** and the non-modified peptide have close elution time in HPLC, posing difficulty in the determination of reaction conversions.

To solve this problem, we change the peptide substrate to its methyl ester (now as peptide **2a** in the revised manuscript). This alteration provides good HPLC separation of the unmodified and modified peptides with other reacting components and make the quantification more reliable. Using peptide methyl ester (**2a**) as the substrate, we have repeated all related experiments and provided updated quantification results. Please see **Supplementary Figures 4-10**.

In these experiments, we did not observe the “degradation products” that appeared in the previous manuscript. The “degradation products” are presumably caused by unknown impurity from the peptide substrate, which caused slow degradation of 1,4-DNIm **1a**.

It is noteworthy that in several assays, we did observe a small portion of 1,4-DNIm **1a** converted into 4-nitroimidazole. In comparison with the stability assay of dinitroimidazole **1a** in HEPES buffer, we propose that the formation of 4-nitroimidazole is promoted by the presence of nucleophilic thiol groups, presumably from non-productive reactions.

Page 3: The authors write “Incubation of SrtA (20 μ M) with compound 4a (200 μ M) for 1 h in PBS buffer, pH 7.0 resulted in quantitative conversion to NBD-modified SrtA, as determined by LC-MS analysis (Fig. 3C).” Based on the provided data it is difficult to determine the conversion. It would be better if the chromatogram would be given with the peaks of the modified and unmodified protein, rather than only the deconvoluted MS spectrum of the modified protein. The MS also does not provide information on the modification site and it can therefore not be concluded that the reagent modifies the active site cysteine of sortase (even though I agree this is very likely). The authors do identify the modification site correctly for BSA.

Response to the comment:

Following the reviewer's requirement, we have provided the MS spectra of SrtA protein, NBD-modified SrtA and RGD-modified SrtA in the updated **Fig. 3C**.

To determine the modification site of NBD-modified SrtA, the sample was digested by trypsin and subjected to LC-MS/MS analysis. Results showed that the modification occurred at Cys192 residue specifically (**Supplementary Figure 17**). In addition, only the modified SrtA segment was observed, whereas the unmodified SrtA segment was not detected, further indicating the completion of the bioconjugation reaction.

Page 3: Even though the gel-labeling results for sortase (Fig 3) are convincing, the authors should perform a competition experiment with iodoacetamide to demonstrate that the reagent indeed binds to the cys residue. Pretreating the protein with this reagent should block labeling if the reaction is indeed selective for cys. (This is an biochemical alternative for identification of the modification site by MS).

Response to the comment:

Thanks for the suggestion. We have followed the reviewer's suggestion to perform additional experiments. Please see **Fig. 3D** for details.

As expected, no fluorescent labeling by NBD-1,4-DNIm was observed in SrtA proteins pre-treated with IAA, indicating that the NBD conjugation of SrtA occurred at Cys residue specifically.

Page 4: Figure 4B: It is difficult to see the mass shift in the MS spectrum. It would be better if a different zoom would be provided.

Response to the comment:

We have provided an updated Fig. 4B with an enlarged scale. The detailed data is also provided in the figure legend.

Due to the quality of BSA commercially purchased, there are some impurity observed by MS.

Page 4: For the stapling of the cys peptides, the authors used ascorbate. I presume this is to prevent disulfide formation, but it is not clearly stated in the text.

Response to the comment:

Thanks for the comment.

Yes, the addition of ascorbate is to keep the cysteine reduced. We have added this information in the main text (Page 3, Left column, late paragraph).

Page 4: The authors conclude “To sum up, under different conditions, 1,4-DNlms selectively react with cysteine thiols or lysine ϵ -amines through two distinct mechanisms, which may account for their vastly different reactivity and chemoselectivity under aqueous and organic conditions.” I do not think this conclusion is fully supported. For example, the authors do not evaluate the reactivity of cysteine residues under organic conditions. Thiols may also react under these conditions. In my opinion, with the here presented data it can therefore not be claimed that 1,4-DNlms react selectively with lysine residues in organic solvent. Either the experiment with a peptide containing a thiol and an amine should be added or the authors should claim that the 1,4-DNlms react selectively with lysines in organic solvents in the absence of thiols.

Response to the comment:

Thanks for the comment.

We agree with the reviewer that the original statement was not accurate enough since the 1,4-DNlms indeed react with thiols very efficiently in organic solvents. Therefore, we have followed the reviewer's suggestion and rephrased this statement (Page 4, Left column, First paragraph).

Supporting information:

The ¹H-NMR of 1b has too many protons. I cannot assign the double triplet at 3.27 ppm. Peaks are lacking in the carbon spectrum.

Response to the comment:

Thanks for the careful examination of the NMR spectrum.

The original ¹H NMR spectra of compound **1b** contains methanol, which accounts for the excess proton signal. We have purified again and removed the solvent properly.

Please see **Supplementary Figure 1** for the new spectrum. In the updated ¹³C NMR spectrum, eight peaks were observed, matching structure of (**1b**).

For compound m3, it is not clear how the regioselectivity is determined. Even though this is the most likely regioisomer, it should still be shown. Furthermore, the HRMS is difficult to read due to the low quality. Finally, it is not clear what is the depicted in the top and the bottom spectrum.

Response to the comment:

There are two isomers for compound **m3**, and HMBC technique was employed to differentiate them. As listed below, in isomer **m3**, a correlation between H9 and C5 is expected, whereas no correlation should exist between H9 and C4. In contrast, in isomer **m3'**, a correlation between H9 and C4 is expected, whereas no correlation should exist between H9 and C5. Our HMBC data show a clear correlation between H9 and C5, but no correlation between H9 and C4 (**Page S43, Supplementary Figure 53**). Therefore, we conclude that the product we obtained is isomer **m3**.

Regarding the HRMS spectra, due to the software of LC-MS in our department, we can only download the screenshot of the HRMS results, which is why the image quality is very low. The top spectrum is the sample spectrum, and the bottom spectrum is the background. To avoid confusion, we have removed these HRMS spectra and provided the HRMS result in the supplementary text along with NMR data of each compound.

Scheme S5: yield of step a) does not match the experimental yield.

Response to the comment:

The yield provided in the legend of **Scheme S5** was wrong and has been corrected to match the experimental description. Please see **Supplementary Figure 62** in the revised supplementary information for detail.

For compound m9: the splitting pattern at 1.35 ppm is strange. I do not expect ddd's for this compound. There are no diastereotopic protons and at most a pentet or double triplet is to be expected for the CH2 at the middle of the linker.

Response to the comment:

We agree with the reviewer that the splitting pattern at 1.35 ppm was wrongly assigned. Based on the NMR spectra, we have changed it to “multipeaks”. Please see Page S55 and **Supplementary Figure 70** in the revised supplementary information for detail.

For compound m11: the splitting pattern for the CH3 of the ester is strange. This should be a triplet.

Response to the comment:

We agree with the reviewer that the splitting pattern was wrongly assigned. Based on the NMR spectra, we have changed it to “triplet”. Please see Page S58 and **Supplementary Figure 74** in the revised supplementary information for detail.

For compound m13: the splitting pattern at 1.39 ppm is strange. I do not expect ddd's for this compound. At most a pentet or double triplet is to be expected for the CH2 at the middle of the linker.

Response to the comment:

We agree with the reviewer that the splitting pattern at 1.39 ppm was wrongly assigned. Based on the NMR spectra, we have changed it to “multiplets”. Please see Page S61 and **Supplementary Figure 78** in the revised supplementary information for detail.

Finally, the authors do not state the previously literature on 1,4-dinitroimidazole compounds. These reagents have been used quite often for the modification of amines in organic solvents.

Response to the comment:

In the revised manuscript, we provide additional experimental data to explore the reaction between 1,4-DNIMs and amines. Selected literatures regarding this reaction were cited. Please see Page 3, Right column, First paragraph, and Reference 38-40.

Some of the early studies, especially from Salwinska group, are not accessible from our database or Web, and therefore are not included.

Reviewer #2 (Remarks to the Author):

I have reviewed the paper and there are items of substance that prevent me from offering enthusiastic support. I feel that the authors must include proposed mechanism in the main text and be in a position to support their claims by robust NMR analysis of regioselectivity in each case. I refer, of course, to the nucleophilic substitution at the imidazole ring. Without robust assignment in each of the cases (whether it is lysine or cysteine nucleophile), it is not possible to make a judgment. The authors do offer some mechanistic schemes in the SI but this is definitely not enough. They need to appear in the main section and have robust support. You just cannot take some obscure chemistry from a Polish journal (which cannot be easily found on the web) and expect that people will simply trust that everything is super clear. No. This is not click chemistry, which was extremely well understood prior to its ultimate application in biology. We are dealing with some fairly tricky (but interesting) reaction sequence in the present case and you just cannot dump this stuff on the readership of Nature Communications and expect everyone to trust you. Please make sure all of the compounds are well characterized.

Response to the comment:

Thanks for the comment.

To address these concerns, we have provided additional experimental data in the revised manuscript regarding the reaction mechanisms. Major revisions include:

1. We have included the proposed reaction mechanisms in the main text (Page 3, Right column) and Figure 1B;

2. The reaction between Cys and 1,4-DNIm **1a** is well characterized, and the structure of product **1b** has been determined by both NMR and crystal structure (Supplementary Figure 1);
3. We agree with the reviewer that the reactions between amines and 1,4-DNIm are less understood based on previous literatures. In this regard, we used ¹⁵N-labeled aniline to react with 1,4-DNIm **1a**, and provided an unambiguous structural assignment to the corresponding product by both NMR and X-ray crystallographic analysis (Fig. 5A, Supplementary figures 19-24). The ¹⁵N of aniline is indeed incorporated in the newly formed imidazole ring.
Through a carefully designed reaction, we also detected a “ring-opening” intermediate (Fig. 5B-C). Although lacking an unambiguous structural determination of the intermediate, our data supports the mechanistic proposal that the reaction follows an ANRORC-like mechanism, consisting of the addition of a nucleophile at C5 position followed by ring-opening and ring closure steps depicted in Fig. 1B. Please see details in the Page 3, Section **“1,4-DNIm modify lysine amine in organic solvents”**.
4. We have cited selected literatures regarding the reactions between amines and 1,4-DNIm. Please see Page 3, Right column, First paragraph, and Reference 38-40. Some of the early studies, especially from Salwinska group, are not accessible from our database or Web, and therefore are not included.

Finally, we have checked the NMR analysis of related compounds to make sure their structures are properly assigned. Please see the revised manuscript.

Reviewer #3 (Remarks to the Author):

This manuscript reports a new method of peptide and protein modification of cysteine and lysine sites. The choice of electrophile for this work is novel as it utilizes a 1,4-dinitroimidazole electrophile, which is, to the best of my knowledge without precedent. The work however needs to be put in more context than is currently presented as perfluoroaryl electrophiles (developed by Pentelute lab) can be used in a similar fashion. The advantage of the method described in this manuscript is smaller and could potentially offer advantages over perfluoroaryl electrophiles.

In order to provide the readership with greater insight, I would suggest the following information be incorporated into a revised manuscript:

1. Compare and contrast the relative advantages and disadvantages of the 1,4-dinitroimidazole reagent versus perfluoroaryl on a model peptide and a protein. This could be conducted by determining the rate constant of the 1,4-dinitroimidazole versus perfluoroaryl versus maleimide for example.

Response to the comment:

Following the reviewer's suggestion, we have synthesized 6,6'-sulfonylbis(1,2,3,4,5-pentafluorobenzene) **pf1**, which is the most reactive perfluoroaryl reagents reported by Pentelute lab, for comparison with 1,4-DNImS. Key conclusions are listed below:

1. Compound **pf1** has poor solubility in aqueous solutions (<0.05 mM), whereas 1,4-DNImS are very soluble (>10 mM). The poor solubility would limit the application of perfluoroaryl reagents in protein

modification. Please see Page 2, Right column, First paragraph for details.

2. Under HEPES buffer, pH 7.4, compound **pf1** is unreactive towards a cysteine-containing tripeptide **2a**, whereas 1,4-DNIm resulted in quantitative conversion. Thus, 1,4-DNIm demonstrate better solubility and reactivity than perfluoroaryl reagents in neutral aqueous solutions. Please see Page 2, Right column, First paragraph for details.
3. The reactivity of compound **pf1** and 1,4-DNIm in organic solvent with a lysine-containing peptide was examined. Results showed that (**pf1**) and 1,4-DNIm **1a** both resulted in full conversion of the peptide substrate in DMSO with DIEA as the base after 30 min. Therefore, 1,4-DNIm has similar reactivity with perfluoroaryl reagent **pf1** for Lys modification. Please see Page 4, Left column, First paragraph and Supplementary Figure 33 for details.

Together, these results show that 1,4-DNIm has advantage in Cys modification under aqueous conditions in comparison with perfluoroaryl reagents. 1,4-DNIm and perfluoroaryl reagents have similar efficiency for Lys modification.

2. Degradation products are observed in the model reaction between 1a and 2a (Fig. 2, Fig. S3). Please identify the degradation products as they appear to be formed in reasonable quantities (e.g., 60s, 120s and 300 s HPLC traces in Fig. 3).

Response to the comment:

Thanks for careful examination of HPLC data.

We notice that in these figures, 1,4-DNIm **1a** and the non-modified peptide have close elution time in HPLC, posing difficulty in the determination of reaction conversions.

To solve this problem, we change the peptide substrate to its methyl ester (now as peptide **2a** in the revised manuscript). This alteration provides good HPLC separation of the unmodified and modified peptides with other reacting components and make the quantification more reliable. Using peptide methyl ester (**2a**) as the substrate, we have repeated all related experiments and provided updated quantification results. Please see Supplementary Figures 4-10.

In these additional experiments, we did not observe the “degradation products” that appeared in the previous manuscript. The “degradation products” are presumably caused by unknown impurity from the peptide substrate, which caused slow degradation of 1,4-DNIm **1a**.

3. The authors mention that the product of maleimide couplings (i.e., thiosuccinimide linkage) is prone to hydrolysis. The authors fail to report the recent works of Caddick, Baker and Chudamsama who have developed hydrolytically stable maleimide and pyridazinedione linkages (Bioconjugate Chem 2018, 29, 486; Chem. Sci. 2016, 7, 799). This work needs to be contextualised against this previous research.

Response to the comment:

Thanks for the suggestion.

We agree with the reviewer that the recent development of maleimide chemistry was not properly included in the previous manuscript. Therefore, we have added related information in this revised manuscript. Please see Page 1 (Right column, Last paragraph), Reference 24-29 for details.

4. The presentation of the reaction schemes is poor. Increase the size of the structures and font in the figures.

Response to the comment:

Following the reviewer’s requirement, we have rearranged the figures for improved quality.

5. Fix geometry of alkyne substituents.

Response to the comment:

The chemical structure of 1,4-DNIm alkyne derivatives have been corrected. Please see Fig. 4.

With the issues addressed, I would consider this work suitable for publication in Nature Commun.

REVIEWERS' COMMENTS:

Reviewer #1 (Remarks to the Author):

The authors have addressed the majority of the comments of the referees:

I am glad to see that by using novel substrates, the analysis of the conversion by LCMS drastically improved.

With the iodoacetamide experiment and the added MS data on sortase unambiguously shows that the cysteine residue is modified.

The comparisons with pentafluoroaryl reagents places the research in a nice perspective. Furthermore, the added references to maleimide chemistry improve the paper.

I appreciate the efforts of the authors to elucidate the mechanisms. One can still argue whether cysteines can attack and perform a retro-Michael, but the presented data with N15 labeled aniline and sterically congested reagents is definitely convincing.

I am also happy that the authors performed additional toxicity studies with the conjugates and demonstrate that the linkages does not have an effect on the viability of the cells. Unfortunately, the authors were not able to assess the immunogenic properties of the conjugate. I do expect most problems here, in particular because of the structural similarities with dinitrophenyl groups, which evoke a strong immune response.

The claim on the selectivity of the reagent improved as well. It remains a bold statement to claim that the reagent selectively modifies cysteines in neutral aqueous conditions. It is truly the case for the model systems that have been tested, but other proteins might react with the reagent. Several catalytic lysine residues are hidden in the binding pocket of proteins and these might react with the reagent. Nonetheless, most of the lysine residues will be non reactive under the aqueous conditions.

Several minor point that remain:

The authors include the MS spectrum from SrtA and modified SrtA, but they do not show the chromatogram. As such it is difficult to determine the conversion, since the conjugate might run differently on the LC MS

The manuscript contains still many typos, especially in the new sections.

Despite this, I do recommend publication.

Reviewer #2 (Remarks to the Author):

The authors have done a very nice job and I support publication. However, if I saw the electron pushing arrows (similar to the kind depicted by the authors in Figure 1B) on a second-year chemistry exam, I would flunk the student who would write something like that. You just cannot do this! How can you possibly have one arrow only, totally neglecting the subsequent chain of events? Please check with your organic chemistry colleagues and they will help.

Reviewer #3 (Remarks to the Author):

The authors of this manuscript have made significant alterations to the main body and conducted additional experiments, which have vastly improved the insight and quality. I would recommend that this manuscript is accepted after the reviewers address the following minor alterations:

Figure 1B. Delete the arrows of the thiol and amine going to the imidazole. As it is drawn, this isn't a mechanistic scheme, rather the authors are trying to point at the site of nucleophilic attack. If this is supposed show a putative reaction mechanism, then curly arrows should be shown throughout the scheme. Otherwise the two curly arrows are confusing.

Supplementary Figure 3. The product of a reaction is not 'HPLC analysis'.

Point-to-point response to reviewers' comments

Responses are highlighted in yellow.

Reviewer #1 (Remarks to the Author):

The authors have addressed the majority of the comments of the referees:

I am glad to see that by using novel substrates, the analysis of the conversion by LCMS drastically improved.

With the iodoacetamide experiment and the added MS data on sortase unambiguously shows that the cysteine residue is modified.

The comparisons with pentafluoroaryl reagents places the research in a nice perspective. Furthermore, the added references to maleimide chemistry improve the paper.

I appreciate the efforts of the authors to elucidate the mechanisms. One can still argue whether cysteines can attack and perform a retro-Michael, but the presented data with N15 labeled aniline and sterically congested reagents is definitely convincing.

I am also happy that the authors performed additional toxicity studies with the conjugates and demonstrate that the linkages does not have an effect on the viability of the cells. Unfortunately, the authors were not able to assess the immunogenic properties of the conjugate. I do expect most problems here, in particular because of the structural similarities with dinitrophenyl groups, which evoke a strong immune response.

The claim on the selectivity of the reagent improved as well. It remains a bold statement to claim that the reagent selectively modifies cysteines in neutral aqueous conditions. It is truly the case for the model systems that have been tested, but other proteins might react with the reagent. Several catalytic lysine residues are hidden in the binding pocket of proteins and these might react with the reagent. Nonetheless, most of the lysine residues will be non reactive under the aqueous conditions.

Several minor point that remain:

The authors include the MS spectrum from SrtA and modified SrtA, but they do not show the chromatogram. As such it is difficult to determine the conversion, since the conjugate might run differently on the LC MS

Response

To address this question, we have provided the TIC and EIC chromatogram derived from the LC-MS analysis of modified SrtA samples, as shown in Supplementary Figure 104 and 106.

From the EIC chromatogram of unmodified and modified SrtA protein in both reactions, we can conclude that the modifications by compounds **4a and **4b** are both near-quantitative. In addition, the modification of compounds **4a** and **4b** did not cause observable difference in the elution time in LC.**

The manuscript contains still many typos, especially in the new sections.

Response

We have carefully examined the manuscript and made corrections to these typos.

Despite this, I do recommend publication.

Reviewer #2 (Remarks to the Author):

The authors have done a very nice job and I support publication. However, if I saw the electron pushing arrows (similar to the kind depicted by the authors in Figure 1B) on a second-year chemistry exam, I would flunk the student who would write something like that. You just cannot do this! How can you possibly have one arrow only, totally neglecting the subsequent chain of events? Please check with your organic chemistry colleagues and they will help.

Response

Thanks for the comment.

We did not intend to draw a formal mechanistic scheme for these two reactions in Figure 1. The arrows only indicate the position of nucleophilic attack by cysteine and lysine.

Since the arrows raise confusion (as also pointed out by Reviewer #3), we have removed the arrows from Figure 1.

Reviewer #3 (Remarks to the Author):

The authors of this manuscript have made significant alterations to the main body and conducted additional experiments, which have vastly improved the insight and quality. I would recommend that this manuscript is accepted after the reviewers address the following minor alterations:

Figure 1B. Delete the arrows of the thiol and amine going to the imidazole. As it is drawn, this isn't a mechanistic scheme, rather the authors are trying to point at the site of nucleophilic attack. If this is supposed show a putative reaction mechanism, then curly arrows should be shown throughout the scheme. Otherwise the two curly arrows are confusing.

Response

Thanks for the comment.

We did not intend to draw a formal mechanistic scheme for these two reactions. The arrows only indicate the position of nucleophilic attack by cysteine and lysine.

Since the arrows raise confusion (as also pointed out by Reviewer #2), we have removed the arrow.

Supplementary Figure 3. The product of a reaction is not 'HPLC analysis'.

Response

Thanks for the comment.

We have made corrections to supplementary figure 3.